# Pre-Endoscopic Scores Predicting Low-Risk Patients with Upper Gastrointestinal Bleeding: A Systematic Review and Meta-Analysis

**DOI:** 10.3390/jcm12165194

**Published:** 2023-08-09

**Authors:** Antoine Boustany, Ali A. Alali, Majid Almadi, Myriam Martel, Alan N. Barkun

**Affiliations:** 1Department of Medicine, Cleveland Clinic Foundation, Cleveland, OH 44195, USA; boustaa@ccf.org; 2Department of Medicine, Faculty of Medicine, Kuwait University, Jabriyah 13110, Kuwait; alialali@moh.gov.kw; 3Department of Medicine, King Saud University, Riyadh 11421, Saudi Arabia; maalmadi@ksu.edu.sa; 4Research Institute of the McGill University Health Center, Montreal, QC H3G 1A4, Canada; myriam.martel@muhc.mcgill.ca; 5Division of Gastroenterology, McGill University Health Center, McGill University, Montréal, QC H3G 1A4, Canada

**Keywords:** upper gastrointestinal bleeding, risk assessment, meta-analysis, glasgow blatchford, rockall, AIMS65, CANUKA, ABC, pre-endoscopic assessment

## Abstract

Background: Several risk scores have attempted to risk stratify patients with acute upper gastrointestinal bleeding (UGIB) who are at a lower risk of requiring hospital-based interventions or negative outcomes including death. This systematic review and meta-analysis aimed to compare predictive abilities of pre-endoscopic scores in prognosticating the absence of adverse events in patients with UGIB. Methods: We searched MEDLINE, EMBASE, Central, and ISI Web of knowledge from inception to February 2023. All fully published studies assessing a pre-endoscopic score in patients with UGIB were included. The primary outcome was a composite score for the need of a hospital-based intervention (endoscopic therapy, surgery, angiography, or blood transfusion). Secondary outcomes included: mortality, rebleeding, or the individual endpoints of the composite outcome. Both proportional and comparative analyses were performed. Results: Thirty-eight studies were included from 2153 citations, (*n* = 36,215 patients). Few patients with a low Glasgow-Blatchford score (GBS) cutoff (0, ≤1 and ≤2) required hospital-based interventions (0.02 (0.01, 0.05), 0.04 (0.02, 0.09) and 0.03 (0.02, 0.07), respectively). The proportions of patients with clinical Rockall (CRS = 0) and ABC (≤3) scores requiring hospital-based intervention were 0.19 (0.15, 0.24) and 0.69 (0.62, 0.75), respectively. GBS (cutoffs 0, ≤1 and ≤2), CRS (cutoffs 0, ≤1 and ≤2), AIMS65 (cutoffs 0 and ≤1) and ABC (cutoffs ≤1 and ≤3) scores all were associated with few patients (0.01–0.04) dying. The proportion of patients suffering other secondary outcomes varied between scoring systems but, in general, was lowest for the GBS. GBS (using cutoffs 0, ≤1 and ≤2) showed excellent discriminative ability in predicting the need for hospital-based interventions (OR 0.02, (0.00, 0.16), 0.00 (0.00, 0.02) and 0.01 (0.00, 0.01), respectively). A CRS cutoff of 0 was less discriminative. For the other secondary outcomes, discriminative abilities varied between scores but, in general, the GBS (using cutoffs up to 2) was clinically useful for most outcomes. Conclusions: A GBS cut-off of one or less prognosticated low-risk patients the best. Expanding the GBS cut-off to 2 maintains prognostic accuracy while allowing more patients to be managed safely as outpatients. The evidence is limited by the number, homogeneity, quality, and generalizability of available data and subjectivity of deciding on clinical impact. Additional, comparative and, ideally, interventional studies are needed.

## 1. Introduction

Acute upper gastrointestinal bleeding (UGIB) is a life-threatening condition that affects one per 1000 population yearly, resulting in more than 300,000 hospital admissions annually in the United States with significant associated costs [1]. Despite the advances in the management of UGIB, it still carries significant morbidity and mortality [2]. However, not all patients with acute UGIB require hospital-based interventions and up to 25% of these patients may successfully be managed on a sole out-patient basis [3]. Therefore, early prediction of negative outcomes among patients with UGIB is crucial to ensure appropriate disposition from the initial point of care. Over the last few decades, several pre-endoscopic risk assessment scores were proposed to risk stratify patients with acute UGIB, including the Glasgow-Blatchford score (GBS) [4], clinical Rockall score (CRS) [5], and AIMS65 score [6]. More recently, the Age, Blood tests and Comorbidities (ABC) [7] and the Canada—United Kingdom—Adelaide (CANUKA) scores [8] were introduced (Table 1). These scores can be used by emergency department or subspecialty physicians when selecting patients with UGIB requiring admission because of a medical, radiological, or surgical intervention.

However, these scoring systems are not routinely used in clinical practice [9], principally due to insufficient validation of their clinical impact in prospective studies. Current practice guidelines for the management of non-variceal UGIB recommend using the GBS to identify low-risk patients with a cutoff of ≤1, but the data supporting this recommendation is quite weak, as reflected by the corresponding very low to low level certainty of evidence using the GRADE rating [10,11]. Therefore, the optimal risk stratification tool for predicting adverse events in a pre-endoscopic setting unfortunately remains unclear [12].

In this systematic review and meta-analysis, we aimed to identify and compare pre-endoscopic published and validated contemporary predictive tools for safely discharging patients with low risk UGIB.

## 2. Materials and Methods

The PICOT question for this study is:Population—patients presenting to the ER with suspected upper GI bleedingIntervention—evaluation of low-risk patient using a pre-endoscopic risk score to predict outcomesControl—non-low risk patients according to varying thresholdsOutcomes—The primary outcome was a composite score for the need of a hospital-based intervention (endoscopic therapy, surgery, angiography, or blood transfusion). Secondary outcomes included: mortality, rebleeding or the individual endpoints of the composite outcomeTime—follow-up up to 30 days from the index bleeding episode

### 2.1. Search Strategy

Systematic searches were performed for full papers and abstracts published up until February 2023 using MEDLINE, EMBASE, Central, and ISI Web of knowledge. Citation selection used a highly sensitive search strategy with Mesh and controlled vocabulary related to (1) UGIB, and (2) pre-endoscopic prognostic scales that are based on pre-endoscopic clinical data. (Appendix A). Recursive searches and cross-referencing were also carried out using a “similar articles” function; hand searches of articles were identified after an initial search.

### 2.2. Study Selection and Patient Population

All fully published studies assessing a pre-endoscopic score in patients with UGIB (including variceal and non-variceal) were included. UGIB was defined as patients presenting with hematemesis, coffee ground vomiting or melena. Exclusion criteria were studies reporting non-human participants, trials not published in English or French, or addressing a pediatric population. In addition, any risk assessment score that was a subsequent modification of an initial publication of a pre-endoscopic score was excluded. The definition of “low-risk” group varied by the individual risk assessment scores. We used the commonly reported score thresholds for determining low-risk patients and varied them to determine the best performing values. Because of the main aim of the trial and the adopted primary outcome (see below), the low-risk group focused more specifically on patients who could be discharged from an emergency room without the performance of an endoscopy at the index visit. This is in keeping with recent guideline recommendations [10,11]. Because of recent guideline recommendations defining a low-risk group as a risk assessment score with ≤1% false negative rate for the outcome of hospital-based intervention or death (e.g., Glasgow-Blatchford score = 0–1), we initially adopted that definition, but also varied the thresholds and risk scores in an attempt to better define their prognostication [11]. Studies that did not specify a cutoff for the low-risk group or did not provide enough data to allow calculation of the low-risk score were excluded.

### 2.3. Validity Assessment

Two reviewers (AB, MM) evaluated the eligibility of all identified citations independently, with a third resolving disagreements (AA). Study quality was assessed using the Ottawa-Newcastle score (NOS) for observational studies [13].

### 2.4. Choice of Outcome

The adopted primary outcome was a previously validated composite score for need of a hospital-based intervention (treatment with transfusion, endoscopic treatment, surgery, or angiography) [14]. This definition was taken from contemporary guidelines as it is specifically tailored to the identification of patients who could be discharged from an emergency room without the performance of an endoscopy at the index visit [10,11]. Secondary outcomes included: rebleeding, mortality, or individual components of the composite outcome. Data will be presented initially as a meta-analysis of proportions (purely descriptive) based on studies that reported outcomes for low-risk patients. We also perform a subsequent meta-analysis assessing studies that included data for both low and greater risk patients allowing for a comparative analysis.

### 2.5. Sensitivity and Subgroup Analyses

Pre-planned possible subgroup and sensitivity analyses for the primary outcome included assessments according to year of publication, quality of studies, performing a fixed rather than a random effect model (when appropriate), and when correcting for double-zero events.

### 2.6. Statistical Analysis and Possible Sources of Statistical Heterogeneity

Categorical estimates of primary and secondary outcomes were reported as proportions and 95% confidence intervals (CI) using weighted random effects models. Continuous variables were reported as means and standard deviations; medians were used if means were not available, and standard deviations (SDs) were calculated or imputed when possible [15]. For comparative studies, effect size was calculated with weighted mean differences (WMDs) for continuous variables. Odds ratios (ORs) were calculated for categorical variables.

The DerSimonian and Laird method [16] for random effect models was applied to all outcomes to determine corresponding overall effect sizes and their confidence intervals. Sensitivity analyses were performed using the Mantel–Haenszel method with fixed effect models when no statistical heterogeneity was noted. WMD were handled as continuous variables using the inverse variance approach. Presence of heterogeneity across studies was defined using a Chi-square test of homogeneity with a 0.10 significance level [15].

The Higgins *I*^2^ statistic [17] was calculated to quantify the proportion of variation in treatment effects attributable to between-study heterogeneity, with values of 25%, 50%, and 75% representing low, moderate, and high heterogeneity, respectively.

For all comparisons, publication bias was evaluated using funnel plots if at least 3 citations were identified. In order to ensure that zero event trials did not significantly affect the heterogeneity or *p*-values, sensitivity analyses were performed where a continuity correction was added to each trial with zero events using the reciprocal of the opposite treatment arm size [18].

All statistical analyses were done using Revman 5.4 and Meta package in R version 2.13.0, (R Foundation for Statistical Computing, Vienna, Austria, 2008).

## 3. Results

### 3.1. Included Studies

Overall, 2153 citations were retrieved; 1497 were rejected based on titles and abstracts, 163 articles were fully reviewed, and 38 studies (*n* = 36,215 patients) were included (PRISMA diagram, Figure 1). Fourteen studies (*n* = 7958 patients) assessed GBS [19,20,21,22,23,24,25,26,27,28,29,30,31], 4 assessed CRS (*n* = 1890 patients) [32,33,34,35], 3 assessed AIMS65 (*n* = 1340 patients) [36,37,38] and 1 study assessed the ABC score (*n* = 2020) [39]. Six studies reported results for both the GBS and CRS (*n* = 2774 patients) [3,40,41,42,43,44], three reported both GBS and AIMS65 (*n* = 1372 patients) [45,46,47] and one assessed GBS and the The Haemoglobin-Urea-Pulse-Systolic blood pressure score (HUPS) (*n* = 934 patients) [48]. The remaining six studies assessed multiple risk scores (*n* = 17,816 patients) [8,14,49,50,51,52]. Table 2 details the included studies. Only scoring systems that had at least three fully published validation studies were included in the results while the others were included only in the Appendix A. Study quality scores using the NOS ranged from 5 to 7 stars out of a possible score of 9, with a mean of 6.6 ± 0.9. Assessing the individual domains of the NOS confirmed the low quality of the studies (Appendix A). No publication bias was observed (data available upon request).

### 3.2. Primary Outcome

The proportion of hospital-based interventions performed (composite outcome) was reported in seven studies (*n* = 4377 patients) [3,19,20,39,40,48,52]. The proportion of low-risk patients requiring hospital-based intervention for GBS cutoffs of 0, ≤1, and ≤2 were 0.02 (0.01, 0.05), 0.04 (0.02, 0.09), and 0.03(0.02, 0.07), respectively. For a CRS cutoff of 0, the proportion was 0.19 (0.15, 0.24), and was 0.69 (0.62, 0.75) for an ABC ≤ 3 (Table 3, Figure 2 and Figure 3). A composite outcome-based analysis was not available for the other scoring systems.

For the comparative analysis between low- and greater-risk groups, data were available from four studies (*n* = 2212 patients) [19,20,39,52]. Scores of GBS = 0 (1 study, *n* = 478 patients) [20], GBS ≤ 1 (1 study, *n* = 569 patients) [20], and GBS ≤ 2 (2 studies, *n* = 998 patients) [19,52] yielded respective ORs of 0.02 (0.00, 0.16), 0.00 (0.00, 0.02) and 0.01 (0.00, 0.04) for predicting hospital-based interventions among low-risk compared to greater-risk groups. A CRS of 0 (1 study, *n* = 478 patients) [52] had an OR of 0.17 (0.08, 0.34), while an ABC ≤ 3 (1 study, *n* = 645 patients) [39] was associated with an OR of 0.42 (0.29, 0.62) (Table 4). Comparative results were not available for the other scoring systems.

### 3.3. Secondary Outcomes

Mortality: Among patients with a GBS of 0 and ≤1, mortality was reported in 0.01 (0.01, 0.03) and 0.01 (0.00, 0.01), respectively. The mortality among patients with a CRS cutoff of 0 was 0.01 (0.00, 0.02), for ≤1 was 0.01(0.00, 0.01) and for ≤2 was 0.02 (0.01, 0.04). For AIMS65 using a cutoff of 0, the mortality was 0.01 (0.01, 0.02), while for an AIMS65 ≤ 1 it was 0.04 (0.03, 0.05). For the ABC score, the proportions for mortality were 0.02 (0.01, 0.12) and 0.10 (0.06, 0.17) for cutoffs of ≤1 and ≤3, respectively. With regard to the comparative analysis, GBS ≤ 1 (OR 0.06 (0.02, 0.20)) and GBS ≤ 2 (OR 0.11 (0.04, 0.27)) had the best predictive ability for the mortality outcome among low-risk compared to greater-risk groups. Detailed results are shown in Table 3 (proportion) and Table 4 (comparative analysis).

Rebleeding: In the proportional analysis, for both GBS and CRS (with cutoffs up to 2) rebleeding occurred in a small proportion of patients identified as low risk (proportions between 0.01 to 0.07) (Table 3). However, in the comparative analysis, only cut-offs of GBS = 0 (OR 0.27 (0.09, 0.97)) and GBS ≤ 1 (OR 0.09 (0.01, 0.68)) were able to discriminate the low-risk from greater-risk groups for the outcome of rebleeding (Table 4).

Blood Transfusion: Blood transfusions were required in 0.01 (0.01, 0.03) and 0.04 (0.03, 0.06) of patients with GBS cutoff of 0 and ≤2, respectively. The proportions for the other scoring systems were higher, as shown in Table 3. For the comparative analysis, GBS using the different cutoffs had the highest predictive ability, as shown in Table 4.

Endoscopic intervention: An endoscopic intervention was required in a small proportion of patients with a low GBS score (GBS = 0, 0.02 (0.01, 0.03), GBS ≤ 1, 0.02 (0.01, 0.02), and GBS ≤ 2, 0.06 (0.04, 0.09)). The proportion of patients identified as low risk using either the CRS or AIMS65 but requiring endoscopic intervention was higher when compared to that using the GBS (Table 3). For the comparative analysis, GBS (using all 3 cutoffs) had the best predictive ability for discriminating low-risk from greater-risk groups for the outcome of endoscopic intervention (Table 4).

Surgical intervention: Patients identified as low risk by GBS, CRS and AIMS65 all had low surgical intervention rates. However, the comparative analysis identified a GBS ≤ 1 and GBS ≤ 2 as the scores with the highest discriminative ability in this regard (OR 0.19 (0.06, 0.60) and OR 0.27 (0.07, 0.97), respectively).

Radiological intervention: Data were only available for the GBS for this outcome. Overall, the proportion of patients requiring radiological intervention was low among GBS = 0 (0.01 (0.00, 0.007)), GBS ≤ 1 (0.00, (0.00, 0.02)) and GBS ≤ 2 (0.01 (0.00, 0.04)). However, GBS did not discriminate well between the low- and greater-risk groups for this endpoint (Table 4).

### 3.4. Sensitivity and Subgroup Analyses

A pre-planned sensitivity analysis according to the year of publication and limiting the assessment to higher quality studies did not alter overall findings (Appendix A).

## 4. Discussion

GIB is the most common cause of hospitalization for GI conditions in the United States, accounting for over half a million admissions annually [54]. Nearly 80% of patients seen in an emergency room with UGIB are admitted to hospital with this condition as principal diagnosis [54]. Yet in over 80% of cases of UGIB, interventions such as endoscopic therapy, blood transfusion or surgery are not needed to stop the bleeding [55]. Although co-morbid conditions may also play a role in the need for hospitalization and other outcomes, not all patients with GIB require admission, hence the critical importance of stratifying patients into being at low or high risk for developing adverse events using validated prognostic scores [56]. A risk assessment tool that correctly identifies very low-risk patients, soon after presentation, who do not need hospital admission or intervention and can be safely discharged to obtain an elective out-patient endoscopy has the potential of reducing health resource utilization in acute UGIB [57].

We focused this systematic review on characterizing and, where possible, looking at the prognostic ability of different scoring schemes in predicting proportions of patients not developing negative outcomes, as well as comparing these amongst patients stratified into low or higher-risk using specific cut-off. We selected for this review scales that can be calculated in the emergency department before any endoscopic intervention (Table 1), thus excluding certain prognostic score assessments such as the PNED [58,59] scale. These needed to have been appropriately validated by sampling cohorts separate from the ones used for development of the individual scale; as well those that should not require endoscopic or in-hospital information, in keeping with our target population of interest. We thus included GBS, CRS, AIMS65 and ABC, but not others that did not fulfill our selection criteria such as the HARBINGER scale [59] (the latter had was reported in less than 3 studies, and did not consider our primary outcome while excluding patients with oozing lesions). More specifically, we assessed prediction of the need for a hospital intervention of any type including endoscopic, surgical, and radiologic therapy, or blood transfusion either individually or as a group (composite outcome measure as proposed by Stanley et al. [14]), as well as the development of rebleeding or mortality. Only the use of the composite outcome measure of avoiding all hospital-based interventions can address the patient population targeted by our meta-analysis since the occurrence of any one of these, even if just one, would increase the risk of discharging the patient from the emergency room without performance of an index endoscopy. This rationale is a very different one than assessing the performance of risk scores in predicting one or many of the hospital-based interventions, and/or rebleeding and/or mortality: all of which relate to patients at higher risk than our target population. Unfortunately, as we did not have patient-level information from the studies, it was impossible for us to identify or report which of the patients who met the composite outcome measure experienced each of its individual components. The continuous outcomes of intensive care unit and hospital lengths of stay were clinically not relevant to the overall focus on outpatient management prediction and were thus not studied.

In the initial part of the meta-analysis, we calculated proportions of patients achieving the various outcomes associated with low-risk allocation for the different scales using optimal cut-offs (summarized in Table 3 with a more complete description included in appendix). The GBS performed well in predicting 0–6% of low-risk patients for all outcomes studied. In contrast, the CRS and AIMS65 were only useful in prognosticating mortality, rebleeding (only the CRS), or surgical intervention (overall 0–7% for low-risk patients).

The aim of this meta-analysis is to compare different thresholds, and thus risk ratios and not absolute test performance characteristics are presented. A meta-analysis of diagnostic tests employs a very different methodology, which was not used as this was not the clinical or methodological aim of our meta-analysis. In the comparative analysis part of our work, only the GBS remained useful (Table 4), and only in predicting the composite outcome of hospital-based interventions and need for blood transfusion as individual outcome. Among the different cut-off values of the GBS that have been assessed in previous studies [3,25,30], a cut-off score of GBS ≤ 1 appeared to be the most discriminative. Indeed, the odds ratios for predicting low-risk patients requiring hospital-based intervention for GBS cutoffs of 0, ≤1, and ≤2 were 0.02 (0.00, 0.16), 0.00 (0.00, 0.02) and 0.01 (0.00, 0.04), respectively. In other words, for example, the likelihood of requiring a hospital-based intervention in a patient with a GBS ≤ 1 (OR = 0.00 (0.00–0.02)) would be, at worst, 50-fold less likely than a patient with a higher GBS score, 95 times out of 100. No CRS or AIMS65 cut-offs were found to be discriminant in the comparative analysis.

The present results provide a more complete review of evidence in support of current guidelines that have suggested the GBS ≤ 1 as a useful score threshold in determining a low risk of adverse events, in turn allowing for safe outpatient management of patients with an early discharge from the emergency room [10,11,60,61]. Interestingly, our results demonstrate that a GBS cut-off score of 2 or less also prognosticated, very accurately, patients at low risk of developing the composite outcome of hospital-based interventions of any type. The potential advantage of adopting this threshold is the greater overall proportion of patients it applies to, and thus can be sent home acutely compared to a smaller number if adopting a GBS threshold of ≤1 (30.5% vs. 19–24%, respectively) [14,24]. This increased applicability needs to be weighed against the minimal additional risk of misclassification. Such trade-off may be very reasonable in a setting of limited resources and could further be assessed using utilities analyses coupled to decision modeling. Importantly, we excluded studies that specifically looked at patients with known co-morbidities or concerning hemodynamic presentation, since these would not be included in any low-risk group as we and the guidelines have defined them. However, patients who initially had neither but may have developed these in time were included in the studies we used to analyze the outcomes (which is in part why there is not a perfect prognostication of patients).

Limitations of the current systematic review include the unavailability of sufficient data to calculate the low-risk prediction performance by some of the scoring system, principally related to an inability to reproduce numerators and denominators from the published information, as defined a priori in our study selection criteria. Additional limitations included the restricted number of studies that could completely inform our systematic review and meta-analysis, and heterogeneity in the selection of patient populations as listed in Table 2, and definitions and selection of individual outcomes. Many scores could not be included as they had not been adequately studied amongst low-risk populations, had not been validated in an independent cohort, or were dependent on endoscopic or hospital-based information such as the HARBINGER scale, qSOFA, shock index, and Progetto Nazionale Emorragia Digestiva (PNED) or an artificial-intelligence-based scoring system [59,62,63,64,65]. There also exists little formal guidance to assess the clinical pertinence of the different prognosticating abilities with regard to balancing the trade-off of accurate outcome prognostication attributable to a provided score threshold versus the proportion of patients that the given cut-off can apply to.

Furthermore, very few studies were interventional in nature, actually assessing the clinical impact of the adoption of the risk score in guiding the downstream clinical management of patients [3]. The prevalence of variceal bleeding amongst all UGIB patients seen in a given practice may also affect the generalizability of the observed results. Indeed, it is important to note that the populations studied typically included patients with both variceal and non-variceal upper gastrointestinal bleeding; although the former usually represent approximately 10% of all acute UGIB [9], depending on local institutional patient mix.

In conclusion, published pre-endoscopic risk scores allow, as a group, good discrimination between populations at low-and higher-risk of developing adverse events. The best performing prognostic scale appears to be the GBS using a cut-off score of 1 or less. Results of our meta-analysis suggest that extending the cut-off to 2 may be reasonable when considering the overall proportion of patients who can be discharged home acutely, potentially allowing for a better utilization of resources. Informing evidence is limited by the number, heterogeneity, quality, and generalizability of the available data. Additional, comparative and, ideally, interventional studies are needed to best confirm these results.

## Figures and Tables

**Figure 1 jcm-12-05194-f001:**
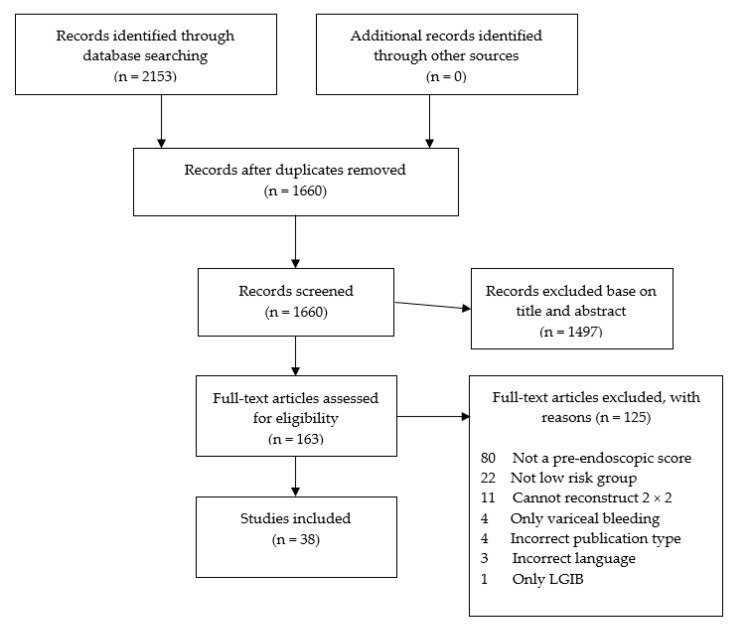
PRISMA diagram.

**Figure 2 jcm-12-05194-f002:**
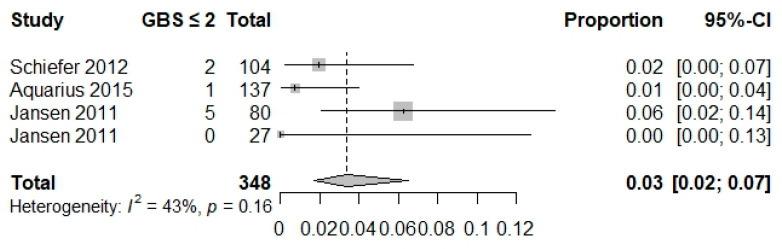
Forest plot composite outcome GBS ≤ 2 [19,48,52].

**Figure 3 jcm-12-05194-f003:**
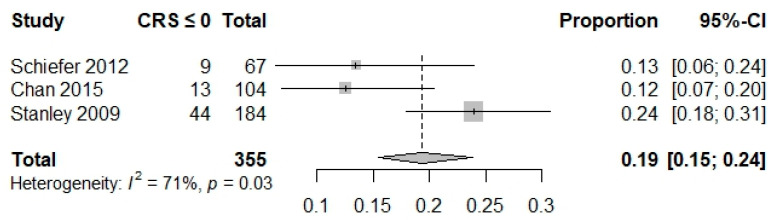
Forest plot composite outcome cRS ≤ 0 [3,40,52].

**Table 1 jcm-12-05194-t001:** Pre-endoscopic risk assessment scores components.

Variables		GBS	CRS	AIMS65	HUPS	ABC	pBBS	pCSMCPI	CANUKA
Urea (mmol/L)	≥6.5<8	2			1				1 (5–9.9)
	≥8<10	3			1				
	≥10<25	4			1				2 (10–14.9) 3 (≥15)
	≥25	6			1				
	>10					1			
Hemoglobin (g/L) for men	<10	6			1				
	≥10<12	3			1				
	≥12<13	1			1				
Hemoglobin (g/L) for women	<100	6			1				
	≥100<120	1			1				
Systolic blood pressure (SPB) (mmHg)	<90	3			1				3 (<80)
	≥90<100	2			1				2 (80–99)
	<100		2						
	≥100<110								
	≥100<120	1							1
Shock	SBP ≥ 100 and HR < 100/min								
	SBP ≥ 100 and HR ≥ 100/min								
Pulse	≥100/min	1			1				1 2 (≥125)
Hemodynamics	Intermediate							1	
	Unstable							2	
ASA class	3					1			
	≥4					3			
Age (years)	≥30<50						1		0
	≥50<60						2		1 (50–64)
	60–69						3		2 (≥65)
	≥60<75					1			
	≥60<80		1						
	≥70						5		
	≥75					2			
	≥80		2						
Albumin (g/dL)	<3					2			
Creatinine (mg/dL)	≥1≤1.5					1			
	>1.5					2			
Time (hours)	<48							1	
	In hospital							2	
Comorbidities	Melena	1							1
	Hematemesis								1
	Syncope	2							1
	Liver disease	2							2
	Liver cirrhosis					2			
	Cardiac failure	2	2						
	Cardiac failure, ischemic heart disease, any major comorbidity		2						
	Renal failure, liver failure, disseminated malignancy		3						
	Altered mental status					2			
	Disseminated malignancy					4			2
	2 comorbidities							1	
	3 comorbidities							2	
	≥4 comorbidities							3	
Illnesses	≥1≤2						1		
	≥3≤4						4		
	≥5						5		
	Acute illness						5		
	Chronic illness						4		

ABC: Age, blood tests and comorbidities; ASA: American Anesthesiology Association; CANUKA: Canada—United Kingdom—Adelaide; CRS: Clinical Rockall score; GBS: Glasgow Blatchford score; HUPS: Hemoglobin–Urea–Pulse–Systolic blood pressure score; pBBS: Pre-endoscopic Baylor Bleeding Score; pCSMCP: Pre-endoscopic Cedars-Sinai Medical Center Predictive Index.

**Table 2 jcm-12-05194-t002:** Details of included studies.

Study (Country), Type of Study, Quality Score Cohorts Size	Cohorts Size	Reported Low Risk Cut-Offs of Pre-Endoscopic Scores Used in the Analysis (Number of Patient with a Score above or below the Threshold)	Definition of Patient Population	Outcome(s) with Extractable Data Used in this Meta-Analysis Definition (When Available)	Definition of UGIB	Definition of Low-Risk Patients
Ak et al., 2021 [36] (Turkey), Retrospective, Score: 7	*n* = 422	AIMS65 ≤ 0: N = 147 AIMS65 > 0: N = 275	Inclusion: patients above the age of 18, who were diagnosed with UGIB and hospitalized after visiting the ED, according to the codes of the International Classification of Diseases (ICD) 10th revision, were included in this study; Exclusion: patients with missing records, patients transferred from other hospitals, patients with variceal bleeding, patients with records of less than 30 days, and patients with a diagnosis other than UGIB after hospitalization were excluded from this study	Mortality 30 days follow-up	Hematemesis, melena, or solid clinical evidence and laboratory support for acute blood loss from the upper GI tract	Defined as the low-risk cut-offs of pre-endoscopic scores reported
Aquarius et al., 2015 [19] (Netherlands), Prospective, Score: 7	*n* = 520	GBS ≤ 2: N = 137 GBS > 2: N = 383	Inclusion: all patients of 18 years or older presenting at the ED for suspected UGIB; Exclusion: NA	Composite outcome: any endoscopic intervention, surgical, or radiological intervention, or need for blood transfusion. Rebleeding Mortality 30 days follow-up	Hematemesis, coffee-ground emesis, and/or melena	Defined as the as low risk cut-offs of pre-endoscopic scores reported
Banister et al., 2018 [20] (UK), Retrospective, Score: 7	*n* = 569	GBS ≤ 1: N = 146 GBS > 1: N = 423	Inclusion: patients aged 18 years or over presenting to the ED or ambulatory care centers with a primary suspected diagnosis of acute UGIB; Exclusion: patients with an inpatient bleed, patients missing information, patients who self-discharged or whether the patient died prior to an assessment being made, if on review of their electronic record they did not have either hematemesis or melaena or if they presented with a chronic GI bleed	Composite outcome: any endoscopic intervention, surgical, or radiological intervention, or need for blood transfusion. Endoscopic therapy Surgery Radiological intervention Blood transfusion Rebleeding Mortality 30 days follow-up	NA	Defined as the as low risk cut-offs of pre-endoscopic scores reported
Bryant et al., 2013 [32] (Australia), Prospective, Score: 6	*n* = 708	cRS ≤ 0: N = 50 cRS > 0: N = NA *	Inclusion: variceal and non-variceal causes of UGIB	Endoscopic therapy 30 days follow-up	Hematemesis (including coffee-ground vomiting) and/ or melena	Defined as the low risk cut-offs of pre-endoscopic scores reported
		cRS ≤ 1: N = 61 cRS > 1: N = NA *				
Chan et al., 2011 [40] (UK), Retrospective, Score: 7	*n* = 432	GBS ≤ 0: N: 40 GBS > 0: N: 392	Inclusion: patients aged 18 years or over presenting to the ED with a primary diagnosis of acute UGIB; Exclusion: patients with an inpatient bleed, lower GI bleeding and who were transferred from another hospital	Composite outcome Endoscopic therapy Radiologic intervention Surgery Rebleeding Mortality 30 days follow-up	Hematemesis including coffee-ground vomiting and/ or melena	Defined as the low risk cut-offs of pre-endoscopic scores reported
		cRS ≤ 0: N: 104 cRS > 0: N: 328				
Chatten et al., 2018 [21] (UK), Retrospective, Score: 7	*n* = 399	GBS ≤ 0: N = 62 GBS > 0: N = 337	Inclusion: patients over the age of 16 who attended the ED or were inpatients with symptoms of an UGIB; Exclusion: patients who did not have an endoscopy	Endoscopic therapy Radiologic intervention Surgery Rebleeding Mortality 30 days follow-up	Hematemesis or melaena	Defined as the as low risk cut-offs of pre-endoscopic scores reported
		GBS ≤ 1: N = 103 GBS > 1: N = 296				
		GBS ≤ 2: N = 136 GBS > 2: N = 263				
Girardin et al., 2014 [22] (Switzerland), Prospectove, Score: 7	*n* = 104	GBS ≤ 0: N: 15 GBS > 0: N: 89	Inclusion: patients over 18 years of age with UGIB; Exclusion: pregnancy and hematochezia	Transfusion Endoscopic therapy Surgery Rebleeding Mortality 30 days follow-up	Hematemesis or coffee ground emesis or with melena	Defined as the as low risk cut-offs of pre-endoscopic scores reported
Gralnek et al., 2004 [41] (USA), Retrospective, Score: 7	*n* = 175	GBS ≤ 0: N: 14 GBS > 0: N: 161	Inclusion: patients over 18 years of age with UGIB according to ICD-9 codes Exclusion: patient who did not undergo endoscopy, developed bleeding while in the hospital, were transferred from another hospital, or bled from a lower-GI source	Rebleeding Mortality	NA	Defined as the as low risk cut-offs of pre-endoscopic scores reported
		cRS ≤ 0: N: 21 cRS > 0: N: 154				
Jansen et al., 2011 [48] (Netherlands), Retrospective, Score: 5	Cohort 1 (*n* = 103)	GBS ≤ 0: N:36 GBS < 0: N:161	Patients had to fulfil all of the following inclusion criteria: (1) presentation at ED with hematemesis, melena, tarry stool or syncope with anemia; (2) diagnosis of acute UGIB was included in the working differential diagnosis formulated by the internist or gastroenterologist; and (3) age over 18 years; Exclusion: patients with signs of chronic bleeding (microcytic anemia)	Need for treatment during the period of 28 days following presentation (blood transfusion, surgical, radiological or endoscopic intervention), rebleeding requiring readmission, or when the patient died	Hematemesis or melena	Defined as the as low risk cut-offs of pre-endoscopic scores reported
		GBS ≤ 1				
		GBS ≤ 2				
	Cohort 2 (*n* = 831)	HUPS ≤ 0: N:14 HUPS > 0: N:817		Composite outcome: any endoscopic intervention, surgical, or radiological intervention, or need for blood transfusion. Surgery Radiological intervention Blood transfusion Rebleeding Mortality 28 days follow-up		
Johnston et al., 2015 [33] (New Zealand), Retrospective, Score: 5	*n* = 388	cRS ≤ 0: N: 42 cRS > 0: N: 346	Inclusion: patients who had a gastroscopy with indication of hematemesis or melena; Exclusion: Outpatients, bleeding during hospitalization, and under 16 years of age	Endoscopic therapy Blood transfusion Surgery Rebleeding Mortality 30 days follow-up for mortality and 14 days follow-up for rebleeding	Hematemesis or melena	Patients were considered low risk if they did not fulfilled any primary or secondary outcomes
Jimenez-Rosales et al., 2023 [49] (Spain), Retrospective, Score: 7	*n* = 795	GBS ≤ 1: N: 27 GBS > 1: N:768	Inclusion: variceal and non-variceal bleed (including inpatient bleed); Exclusion: refusal to sign the informed consent	Mortality 30 days follow-up	Melena and/or hematemesis (including coffee ground vomiting)	Defined as the low risk cut-offs of pre-endoscopic scores reported
		AIMS65 ≤ 1: N: 477 AIMS65 > 1: N: 318				
		ABC ≤ 1: N: 334 ABC > 1: N: 461				
Kayali et al., 2017 [23] (Turkey), Retrospective, Score: 6	*n* = 188	GBS ≤ 2: N: 9 GBS > 2: N: 179	Inclusion: patients aged above 18 with UGIB complaints; Exclusion: NA	Mortality Follow-up: NA	NA	Defined as the as low risk cut-offs of pre-endoscopic scores reported
Kherad et al., 2022 [39] (Canada), Retrospective, Score: 7	*n* = 645	ABC ≤ 3: N: 228 ABC > 3: N: 417	Inclusion: all hospitalized patients of at least 18 years of age with a primary or secondary discharge diagnosis of nonvariceal and variceal UGIB using ICD-9 and ICD-10 codes Exclusion: Outpatients and transfers from other hospitals	Composite outcome: any endoscopic intervention, surgical, or angiography, or need for blood transfusion. Blood transfusion Rebleeding Mortality 30 days follow-up	Melena and/or hematemesis	Defined as the low-risk cut-offs of pre-endoscopic scores reported
Lahiff et al., 2012 [42] (Ireland), Retrospective, Score: 7	*n* = 200	GBS ≤ 0: N: 21 GBS > 0: N: 179	Inclusion: NA; Exclusion: patients with chronic anaemia, those with a lower GI source for bleeding and endoscopies performed for suspected UGIB for in-patients	Endoscopic intervention Rebleeding Mortality 30 days follow-up	Hematemesis (fresh blood or coffee-ground emesis), melena and hematochezia	Defined as the as low risk cut-offs of pre-endoscopic scores reported
		GBS ≤ 2: N: 57 GBS > 2: N: 143				
		cRS ≤ 0: N: 42 cRS > 0: N: 158				
Laursen et al., 2012 [50] (Denmark), Prospective, Score: 7	*n* = 831	GBS ≤ 0: N: 96 GBS > 0: N: 735	Inclusion: patients presenting with UGIB while already admitted for other reasons. Exclusion: Patients with UGIB transferred from other hospitals	Endoscopic intervention Mortality 30 days follow-up	Hematemesis, coffee-ground vomit, or melena	Defined as patients who did not need hospital-based intervention and survived more than 30 days from day of admission
		GBS ≤ 2: N: 173 GBS > 2: N: 658				
		cRS ≤ 0: N: 130 cRS > 0: N: 701				
		pBBS ≤ 0: N: 26 pBBS > 0: N:805				
		pBBS ≤ 1: N:86 pBBS > 1: N: 745				
		pBBS ≤ 2: N: 140 pBBS > 2: N:691				
		pCSMCPI ≤ 0: N: 26 pCSMCPI > 0: N: 805				
Laursen et al., 2015 [24] (Denmark), Prospective, Score: 7	*n* = 2305	GBS ≤ 0: N: 313 GBS > 0: NA *	Inclusion: Patients with UGIB; Exclusion: patients experiencing UGIB while already inpatients for another reason	Transfusion Mortality 30 days follow-up	Hematemesis, coffee-ground vomit, or melena	Defined as patients who did not need a blood transfusion or hemostatic intervention, and did not die during the index admission
		GBS ≤ 1: N: 562 GBS > 1: NA *				
		GBS ≤ 2: N: 704 GBS > 2: NA *				
Leiman et al., 2017 [25] (USA), Retrospective, Score: 5	N = 66	GBS ≤ 0: N: 66 GBS > 0: NA *	Inclusion: Diagnosis of UGIB; Exclusion: under age 18, did not report symptoms of UGIB or had a GBS of 1 or more, and those with vital sign or laboratory abnormalities that would preclude them from being low risk	Endoscopic therapy Surgery Blood transfusion Mortality Follow-up: NA	Hematemesis or coffee ground emesis	Defined as the as low risk cut-offs of pre-endoscopic scores reported
Lima et al., 2013 [34] (Brazil), Prospective, Score: 7	*n* = 656	cRS ≤ 0: N: 94 cRS > 0: N: 562	Inclusion: clinical evidence of UGIB or a history of hematemesis, coffee ground vomiting or melena within 24 h preceding the admission; Exclusion: Bleeding from varices or portal hypertensive gastropathy	Rebleeding Mortality 30 days follow-up	Hematemesis, coffee ground vomiting or melena	Defined as the low risk cut-offs of pre-endoscopic scores reported
		cRS ≤ 1: N: 227 cRS > 1: N: 429				
		cRS ≤ 2: N: 360 cRS > 2: N: 296				
Lu et al., 2020 [37] (China), Retrospective, Score: 6	*n* = 284	AIMS65 ≤ 1: N: 200 AIMS65 > 1: N: 84	Inclusion: patients hospitalized within 48 hours of endoscopy and diagnosed with UGIB; Exclusion: (1) insufficient laboratory data for calculating the risk scores; (2) endoscopic examination not performed; (3) hemorrhage other than UGIB; (4) unacceptable specification system treatment, including automatic discharge and transfer of patients; and (5) non-acute UGIB cause death	Mortality Follow-up: NA	NA	Defined as the low risk cut-offs of pre-endoscopic scores reported
Matsuhashi et al., 2021 [51] (Japan), Retrospective, Score: 6	Cohort 1 (*n* = 1,380)	GBS ≤ 1: N: 10 GBS < 1: N: 1370	Inclusion: patients with non-variceal UGIB; Exclusion: (1) bleeding from malignancy and (2) bleeding after endoscopic resection	Mortality Follow-up: NA	Hematemesis, coffee ground vomiting or melena	Defined as the low risk cut-offs of pre-endoscopic scores reported
		cRS ≤ 0: N: 129 cRS > 1: N: 1251				
		AIMS65 ≤ 1: N: 620 AIMS65 > 1: N: 760				
		ABC ≤ 3: N: 619 ABC > 3: N: 761				
	Cohort 2 (*n* = 825)	GBS ≤ 1: N: 15 GBS < 1: N: 810				
		cRS ≤ 0: N: 67 cRS > 1: N: 758				
		AIMS65 ≤ 1: N: 342 AIMS65 > 1: N: 483				
		ABC ≤ 3: N: 326 ABC > 3: N: 499				
Meltzer et al., 2013 [43] (USA), Retrospective, Score: 6	*n* = 690	GBS ≤ 0: N: 63 GBS > 0: N: 627	Inclusion: Patients aged 18 years or older and ED final diagnoses of GI bleed (unspecified) or UGIB (any cause); Exclusion: NA	Endoscopic therapy Follow-up: NA	Hematemesis, coffee ground vomiting or melena	Defined as the as low risk cut-offs of pre-endoscopic scores reported
		cRS ≤ 0: N: 122 cRS > 0: N: 568				
Mustafa et al., 2015 [26] (UK), Prospective, Score: 7	*n* = 514	GBS ≤ 1: N: 183 GBS > 1: N: 331	NA	Composite outcome: blood transfusion, endoscopic treatment, radiological intervention or surgery Endoscopic therapy Radiologic intervention Blood transfusion Surgery Mortality Follow-up: 30 days	Hematemesis, coffee-ground vomit or melena	Defined as the as low risk cut-offs of pre-endoscopic scores reported
Oakland et al., 2019 [8] (UK), Retrospective, Score: 6	*n* = 1606	GBS ≤ 0: N: 187 GBS > 0: N: 1419	Inclusion: only patients for whom all 3 risk scores (GBS, cRS, CANUKA) could be calculated were included; Exclusion: patients with missing data on any of the variables used to derive the 3 risk scores were excluded	Endoscopic therapy Blood transfusion Rebleeding Mortality Follow-up: 30 days	NA	Defined clinically as patients who did not require (or experience) any of the following: RBC transfusion, rebleeding, therapeutic endoscopy, interventional radiology or surgery, or mortality.
		GBS ≤ 1: N: 381 GBS > 1: N: 1225				
		cRS ≤ 0: N: 329 cRS > 0: N: 1277				
		cRS ≤ 1: N: 605 cRS > 1: N: 1001				
		CANUKA ≤ 0: N: 9 CANUKA > 0: N: 1597				
		CANUKA ≤ 1: N: 109 CANUKA > 1: N: 1497				
Pang et al., 2010 [27] (USA), Prospective, Score: 7	*n* = 1087	GBS ≤ 0: N: 50 GBS > 0: N: 1037	Inclusion: Patients with UGIB; Exclusion: patients younger than the age of 18 years and those with primary diagnoses other than UGIB were excluded from the study	Endoscopic therapy Rebleeding Mortality Follow-up: 30 days	Hematemesis, coffee grounds vomiting, melena, or hematochezia	Defined as the as low risk cut-offs of pre-endoscopic scores reported
Park et al., 2015 [38] (South Korea), Retrospective, Score: 7	*n* = 634	AIMS65 ≤ 1: N: 434 AIMS65 > 1: N: 200	Inclusion: Any adult (age 18 or older) with any UGIB Exclusion: upper endoscopy not performed, lower or small bowel bleed, variceal bleed, cancer- or post-procedure bleed	Mortality Follow-up: NA	Melena, hematemesis and/or hematochezia	Defined as the low-risk cut-offs of pre-endoscopic scores reported
Robins et al., 2007 [28] (UK), Retrospective, Score: 7	*n* = 194	GBS ≤ 1: N: 194 GBS > 1: N: NA	Inclusion: Patients with UGIB Exclusion: age greater than 60 years, postural fall in systolic blood pressure greater than 20 mmHg, known esophageal varices, receiving anticoagulation, and social circumstances that prevent discharge within 24 h	Endoscopic therapy Blood transfusion Follow-up: 30 days	NA	Defined as the as low risk cut-offs of pre-endoscopic scores reported
Ryan et al., 2021 [31] (Australia), Retrospective, Score: 7	*n* = 181	GBS ≤2: N: 49 GBS > 2: N: 132	Inclusion: Patients with UGIB and a GBS was able to be calculated; Exclusion: patients presenting with iron deficiency anemia without evidence of UGIB, confirmation that the source of bleeding was not from the upper GI tract (e.g., oropharynx or lower GI tract) and patients with incomplete data to calculate a GBS	Endoscopic therapy Radiologic intervention Surgery Rebleeding Mortality Follow-up: 30 days	Hematemesis, coffee grounds vomiting, or melena,	Defined as the low-risk cut-offs of pre-endoscopic scores reported
Samreen et al., 2016 [29] (Pakistan), Retrospective, Score: 6	*n* = 280	GBS ≤2: N: 51 GBS > 2: N: 229	Inclusion: patients age > 18 year old presenting to the ED with UGIB of any cause (variceal or non-variceal); Exclusion: patients with age < 18 years and those not admitted through emergency were excluded	Endoscopic therapy Follow-up: NA	Hematemesis, melena or bloody nasogastric tube aspirate	Defined as the as low risk cut-offs of pre-endoscopic scores reported
Sasaki et al., 2022 [45] (Japan), Retrospective, Score: 7	*n* = 675	GBS ≤ 1: N: 39 GBS > 1: N: 636	Inclusion: patients with suspected non-variceal UGIB	Endoscopic therapy	Hematemesis, black stool, syncope, and anemia	Defined as the low-risk cut-offs of pre-endoscopic scores reported
		AIMS65 ≤ 1: N: 312 AIMS65 > 1: N: 363				
Schiefer et al., 2012 [52] (Netherlands), Retrospective, Score: 7	*n* = 478	GBS ≤ 0: N: 39 GBS > 0: N: 439	Inclusion: all patients presenting to the ED with suspected UGIB; Exclusion: symptomatic anemia from chronic GI bleeding, and under 18 years of age	Composite outcome: endoscopic therapy, surgical or radiological treatment, receiving blood transfusion. Endoscopic therapy Follow-up: 28 days	Hematemesis or melena, or unexplained acute drop in hemoglobin level	Defined as the low-risk cut-offs of pre-endoscopic scores reported
		GBS ≤ 2: N: 104 GBS > 2: N: 374				
		cRS ≤ 0: N: 67 cRS > 0: N: 411				
		HUPS ≤ 0: N: 56 HUPS > 0: N: 422				
Shrestha et al., 2014 [44] (Nepal), Prospective, Score: 7	*n* = 589	GBS ≤ 0: N: 12 GBS > 0: N: 577	Inclusion: all UGIB patients from both inpatients and outpatients; Exclusion: patients presenting with chronic anemia and those with a lower GI source for bleeding	Blood transfusion Surgery Rebleeding Mortality Follow-up: 30 days	Hematemesis, melena, nasogastric aspirate containing blood and hematochezia caused by the blood loss from the upper GI tract.	Defined as the low-risk cut-offs of pre-endoscopic scores reported
		GBS ≤ 1: N: 42 GBS > 1: N: 547				
		GBS ≤ 2: N: 76 GBS > 2: N: 513				
		cRS ≤ 0: N: 122 cRS > 0: N: 467				
		cRS ≤ 1: N: 203 cRS > 1: N: 386				
		cRS ≤ 2: N: 320 cRS >2: N: 269				
Stanley et al., 2009 [3] (UK), Prospective, Score: 7	Cohort 1 (*n* = 676)	GBS ≤ 0: N: 105 GBS > 0: N: 551	Inclusion: patients with UGIB; Exclusion: inpatients with UGIB Exclusion: NA	Composite outcome: blood transfusion, endoscopic treatment, or surgery Endoscopic therapy Blood transfusion Surgery Mortality Follow-up: 6 months	Defined as hematemesis coffee-ground vomit, or melena	Defined as the as low risk cut-offs of pre-endoscopic scores reported
		cRS ≤ 0: N: 184 cRS > 0: N: 492				
	Cohort 2 (*n* = 572)	GBS ≤ 0: N: 123 GBS > 0: N: 449				
Stanley et al., 2017 [14] (UK), Prospective, Score: 7	*n* = 2868	GBS ≤ 1: N: 564 GBS > 1: N: 2304	Inclusion: patients with UGIB; Exclusion: patients who developed UGIB while an inpatient for another reason	Endoscopic therapy Blood transfusion Mortality Follow-up: 30 days	Hematemesis, coffee-ground vomiting, or melena	Defined as the low-risk cut-offs of pre-endoscopic scores reported
		cRS ≤ 0: N: 436 cRS > 0: N: 2432				
		AIMS65 ≤ 0: N: 865 AIMS65 > 0: N: 2003				
Stephens et al., 2009 [30] (UK), Prospective, Score: 7	Cohort 1 (*n* = 232)	GBS ≤ 0: N: 29 GBS > 0: N: 203	Inclusion: patients with UGIB; Exclusion: patients who have UGIB while an inpatient in hospital for another cause and those home alone whatever their GBS was	Endoscopic therapy Blood transfusion Surgery Mortality Follow-up: 4 -6 weeks	Hematemesis (including coffee ground vomiting) and/or melena	Patients with ‘low-risk’ UGIB fulfilling the above criteria were considered for management in the community
		GBS ≤ 1: N: 53 GBS > 1: N: 179				
		GBS ≤ 2: N: 66 GBS > 2: N: 166				
	Cohort 2 (*n* = 304)	GBS ≤ 0: N: 46 GBS > 0: N: 258	Inclusion: patients with UGIB and GBS ≤2 and age <70 years, were accompanied at home; had a telephone and transport; had no active significant comorbidities; were not taking warfarin and did not have suspected variceal bleeding; Exclusion: NA			
		GBS ≤ 1: N: 93 GBS > 1: N: 211				
		GBS ≤ 2: N: 123 GBS > 2: N: 181				
Tham et al., 2006 [35] (UK), Retrospective, Score: 6	*n* = 102	cRS ≤ 0: N: 38 cRS > 0: N: 64	Inclusion: acute non-variceal UGIB were identified using ICD-9 codes; Exclusion: NA	Blood transfusion Surgery Rebleeding Mortality Follow-up: NA	NA	A clinical Rockall Score of 0 was considered ‘‘low risk’’ for adverse outcomes (recurrent bleeding and mortality) related to acute upper gastrointestinal hemorrhage
		cRS ≤ 1: N: 51 cRS > 1: N: 51				
		cRS ≤ 2: N: 67 cRS > 2: N: 35				
Thanapirom et al., 2012 [53] (Thailand), Prospective, Score: 6	*n* = 756	GBS ≤ 2: N: 99 GBS > 2: N: 657	Inclusion: Patients with UGIB; Exclusion: patients who refused endoscopic examination	Rebleeding Mortality Follow-up: 30 days	Hematemesis (including coffee-ground vomiting), melena, and hematochezia	Defined as the low-risk cut-offs of pre-endoscopic scores reported
Yaka et al., 2015 [46] (Turkey), Prospective, Score: 7	*n* = 254	GBS ≤ 0: N: 16 GBS > 0: N: 238	Inclusion: Adult patients with UGIB (variceal and non-variceal) Adult; Exclusion: Patients who received any treatment at another institution, visited the ED due to a rebleeding episode from a prior upper GI bleeding, had incomplete data for score calculation or outcome determination, or whose source of bleeding was the lower GI tract	Endoscopic therapy Blood transfusion Surgery Follow-up: 30 days	Hematemesis, “coffee-ground” vomit, melena, hematochezia	Patients who did not require blood transfusions or suffer the composite clinical outcomes were considered low-risk patients.
		GBS ≤ 2: N: 48 GBS > 2: N: 206				
		AIMS65 ≤ 0: N: 101 AIMS65 > 0: N: 153				
Zhong et al., 2016 [47] (China), Prospective, Score: 6	Cohort 1 (*n* = 320)	GBS ≤ 2: N: 101 GBS > 2: N: 219	Inclusion: patients aged >18 years who were admitted with acute UGIB; Exclusion: (i) recurrent episode of acute UGIB during the study period; (ii) patients diagnosed as non-acute UGIB (iii) patients with incomplete emergency medical data for the calculation of GBS, and AIMS65 score; (iv) patients who had received treatments at other institutions; (v) patients in whom the bleeding source was confirmed to be the lower GI tract	Rebleeding Mortality Follow-up: NA	Hematemesis, coffee- ground vomitus, melena, and/or hematochezia	Defined as the low-risk cut-offs of pre-endoscopic scores reported
		AIMS65 ≤ 1: N: 134 AIMS65 > 1: N: 136				

ABC: Age, blood tests and comorbidities; ED: Emergency department; NA: Not available; UGIB: Upper gastrointestinal bleed; GBS: Glasgow Blatchford Score; cRS: Clinical Rockall score; LOS: Length of stay; ICU: Intensive care unit; pBBS: Pre-endoscopic Baylor Bleeding Score; pCSMCP: Pre-endoscopic Cedars-Sinai Medical Center Predictive Index; HUPS: Hemoglobin–Urea–Pulse–Systolic blood pressure score; CANUKA: Canada—United Kingdom—Adelaide; * Study only used for proportion meta-analysis.

**Table 3 jcm-12-05194-t003:** Primary and secondary outcomes for risk assessment scores (expressed as proportions).

	GBS Proportion (95% CI)			CRS Proportions (95% CI)			AIMS65 Proportions (95% CI)		ABC Proportions (95% CI)	
	0	≤1	≤2	0	≤1	≤2	0	≤1	≤1	≤3
Composite outcome	0.02 (0.01–0.05) I^2^ = 0% Studies = 4 N = 347	0.04 (0.02–0.09) I^2^ = 49% Studies = 2 N = 237	0.03 (0.02–0.07) I^2^ = 43% Studies = 3 N = 348	0.19 (0.15–0.24) I^2^ = 71% Studies = 3 N = 355	NA	NA	NA	NA	NA	0.69 (0.62; 0.75) I^2^ = NA Studies = 1 N = 228
Mortality	(0.01–0.03) I^2^ = 0% Studies = 14 N = 1200	(0.00–0.01) I^2^ = 0% Studies = 10 N = 2179	NA	(0.00–0.02) I^2^ = 0% Studies = 11 N = 1634	(0.00–0.01) I^2^ = 0% Studies = 4 N = 1086	0.02 (0.01–0.04) I^2^ = 0% Studies = 3 N = 747	(0.01–0.02) I^2^ = 78% Studies = 2 N = 1012	0.04 (0.03–0.05) I^2^ = 89% Studies = 7 N = 2788	0.30 (0.25; 0.36) I^2^ = NA Studies = 1 N = 312	0.02 (0.01; 0.03) I^2^ = 73% Studies = 2 N = 1173
Rebleeding	0.02 (0.01–0.04) I^2^ = 0% Studies = 9 N = 428	(0.01–0.03) I^2^ = 0% Studies = 4 N = 672	0.03 (0.02–0.06) I^2^ = 52% Studies = 8 N = 703	0.04 (0.03–0.06) I^2^ = 46% Studies = 7 N = 688	0.05 (0.04–0.06) I^2^ = 67% Studies = 4 N = 1086	0.07 (0.05–0.09) I^2^ = 67% Studies = 3 N = 747	NA	0.31 (0.24–0.39) I^2^ = 98% Studies = 2 N = 281	NA	0.11 (0.07; 0.15) I^2^ = NA Studies = 1 N = 228
Blood transfusion	(0.01–0.03) I^2^ = 0% Studies = 8 N = 923	NA	0.04 (0.03–0.06) I^2^ = 88% Studies = 4 N = 1037	0.14 (0.12–0.16) I^2^ = 86% Studies = 7 N = 1255	0.22 (0.20–0.25) I^2^ = 95% Studies = 3 N = 859	0.48 (0.43–0.54) I^2^ = 97% Studies = 2 N = 387	0.20 (0.18–0.23) I^2^ = 70% Studies = 2 N = 966	NA	NA	0.60 (0.53; 0.66) I^2^ = NA Studies = 1 N = 228
Endoscopic intervention	0.02 (0.01–0.03) I^2^ = 0% Studies = 13 N = 969	0.02 (0.01–0.02) I^2^ = 0% Studies = 8 N = 1756	0.06 (0.04–0.09) I^2^ = 76% Studies = 8 N = 807	0.08 (0.07–0.10) I^2^ = 0% Studies = 8 N = 1255	0.15 (0.12–0.18) I^2^ = 94% Studies = 2 N = 666	NA	0.12 (0.10–0.15) I^2^ = 0% Studies = 2 N = 966	0.30 (0.25–0.36) I^2^ = NA (1 study) Study = 1 N = 312	NA	NA
Surgical intervention	(0.01–0.03) I^2^ = 0% Studies = 8 N = 525	(0.00–0.02) I^2^ = 0% Studies = 5 N = 620	(0.00–0.02) I^2^ = 0% Studies = 5 N = 498	(0.00–0.03) I^2^ = 0% Studies = 4 N = 306	(0.00–0.03) I^2^ = 0% Studies = 2 N = 254	(0.00–0.02) I^2^ = 0% Studies = 2 N = 387	(0.00–0.04) I^2^ = NA (1 study) Study = 1 N = 101	NA	NA	NA
Radiological intervention	0.01 (0.00–0.07) I^2^ = 0% Studies = 2 N = 102	0.00(0.00–0.02) I^2^ = 0% Studies = 3 N = 432	0.01 (0.00–0.04) I^2^ = 0% Studies = 2 N = 185	NA	NA	NA	NA	NA	NA	NA

Results expressed as proportions (95% confidence interval); NA: Not available.

**Table 4 jcm-12-05194-t004:** Primary and secondary outcomes for risk assessment scores comparing low-risk to higher-risk patients (expressed as odds ratio).

	GBS OR (95% CI)			CRS OR (95% CI)			AIMS65 OR (95% CI)		ABC OR (95% CI)
Outcomes	0	≤1	≤2	0	≤1	≤2	0	≤1	≤3
Composite outcome	0.02 (0.00–0.16) Studies = 1 (*n* = 478)	0.00 (0.00–0.02) Studies = 1 (*n* = 569)	0.01 (0.00–0.04) Studies = 2 (*n* = 1102)	0.17 (0.08–0.34) Studies = 1 (*n* = 545)	NR	NR	NR	NR	0.42 (0.29, 0.62) Studies = 1 (*n* = 645)
Mortality	0.27 (0.09–0.97) Studies = 7 (*n* = 1880)	0.06 (0.02–0.20) Studies = 9 (*n* = 5607)	0.11 (0.04–0.27) Studies = 8 (*n* = 2467)	0.18 (0.08–0.43) Studies = 6 (*n* = 3940)	0.13 (0.05–0.34) Studies = 3 (*n* = 1347)	0.28 (0.16–0.50) Studies = 3 (*n* = 1347)	0.27 (0.09–0.78) Studies = 1 (*n* = 422)	0.13 (0.09–0.18) Studies = 7 (*n* = 4430)	0.10 (0.06; 0.17) Studies = 2 (*n* = 2750)
Rebleeding	0.26 (0.07–0.91) Studies = 5 (*n* = 1450)	0.09 (0.01–0.68) Studies = 2 (*n* = 988)	0.24 (0.05–1.01) Studies = 4 (*n* = 1189)	0.41 (0.10–1.63) Studies = 4 (*n* = 1735)	0.71 (0.45–1.11) Studies = 3 (*n* = 1347)	0.50 (0.15–1.67) Studies = 3 (*n* = 1347)	NR	0.30 (0.04–2.37) Studies = 2 (*n* = 512)	0.47 (0.29; 0.79) Studies = 1 (*n* = 645)
Blood transfusion	0.03 (0.01–0.08) Studies = 6 (*n* = 1587)	0.01 (0.00–0.03) Studies = 5 (*n* = 2208)	0.01 (0.01–0.04) Studies = 3 (*n* = 810)	0.28 (0.19–0.40) Studies = 3 (*n* = 1079)	0.16 (0.03–0.80) Studies = 2 (*n* = 691)	0.30 (0.04–2.13) Studies = 2 (*n* = 691)	0.12 (0.07–0.21) Studies = 1 (*n* = 254)	NR	0.47 (0.33; 0.66) Studies = 1 (*n* = 645)
Endoscopic intervention	0.04 (0.02–0.11) Studies = 7 (*n* = 2484)	0.02 (0.01–0.05) Studies = 6 (*n* = 2693)	0.07 (0.04–0.13) Studies = 6 (*n* = 1650)	0.53 (0.20–1.41) Studies = 1 (*n* = 388)	NR	NR	0.48 (0.24–0.96) Studies = 1 (*n* = 254)	0.78 (0.57–1.08) Studies = 1 (*n* = 675)	NR
Surgical intervention	0.93 (0.30–2.86) Studies = 7 (*n* = 1986)	0.19 (0.06–0.60) Studies = 6 (*n* = 2607)	0.27 (0.07–0.97) Studies = 5 (*n* = 1778)	0.37 (0.07–2.05) Studies = 3 (*n* = 1079)	0.17 (0.02–1.42) Studies = 2 (*n* = 691)	0.29 (0.07–1.25) Studies = 2 (*n* = 691)	0.13 (0.01–2.43) Studies = 1 (*n* = 254)	NR	NR
Radiological intervention	1.07 (0.05–22.6) Studies = 1 (*n* = 399)	0.20 (0.04–1.10) Studies = 3 (*n* = 1482)	0.38 (0.02–8.04) Studies = 1 (*n* = 399)	NR	NR	NR	NR	NR	NR

Results expressed as odds ratios (95% confidence intervals) demonstrating the likelihood of achieving outcomes amongst patient with a risk score below versus above the given threshold. NR: Not reported.

## Data Availability

The data presented in this study are available on request from the corresponding author.

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
