# Peer review of "Pre-Endoscopic Scores Predicting Low-Risk Patients with Upper Gastrointestinal Bleeding: A Systematic Review and Meta-Analysis"

_jcm, 2023, doi:10.3390/jcm12165194_

Round 1

Reviewer 1 Report

Authors showed that a GBS cut-off of one or less prognosticated low-risk patients best.

Although the aim of this study is interesting, authors should concern below points.

1.     The percentage of each outcome in the composite outcome should be indicated.

Of this composite outcome, the majority may be blood transfusion, which is virtually equivalent to predicting blood transfusion. This point should be discussed.

2.     The sensitivity and specificity for each cutoff value should be summarized. It would also be of interest how many people below that cutoff are included in the total. If the sensitivity is high but the number of people in the population below the cutoff is small, it will not be possible to select many people who do not need treatment.

3.     No mention is made in the discussion of the recently reported HARBINGER to determine the indication for emergency endoscopy. (Gastrointest Endosc. 2020 Sep;92(3):578-588.e4.) The authors should mention it in the discussion.

Also, the primary outcome is high risk stigmata, i.e., need for endoscopic treatment, and the cutoffs for GBS and AIMS65 to rule out are both reported as 1. The authors should include this paper in the meta-analysis.

Author Response

1. The percentage of each outcome in the composite outcome should be indicated.

Of this composite outcome, the majority may be blood transfusion, which is virtually equivalent to predicting blood transfusion. This point should be discussed.

The individual components of the composite outcome were reported as secondary outcomes when available. In addition, the composite outcome (hospital-based intervention) aims at detecting any intervention which warrants admission regardless of its perceived seriousness. So in case of blood transfusion, even if all the composite outcome was composed entirely of blood transfusion outcome, this would be an indication for admission which is the main aim of the risk assessment scores addressing low-risk patients (i.e.. Identify patients who are unlikely to require hospital-based intervention and be managed as outpatient safely). Furthermore, the choice of this composite outcome has been previously validated and endorsed by guidelines, justifying its choice.

2. The sensitivity and specificity for each cutoff value should be summarized. It would also be of interest how many people below that cutoff are included in the total. If the sensitivity is high but the number of people in the population below the cutoff is small, it will not be possible to select many people who do not need treatment.

We have provided these data and such a discussion for the only risk measure in which the data are sufficient to explore such a trade-off with sufficient statistical power. It is the GBS in which we clearly discuss in the results and the discussion the trade-off of test performance characteristics with the proportion of patients affected by using a threshold score of 1 or 2.

3. No mention is made in the discussion of the recently reported HARBINGER to determine the indication for emergency endoscopy. (Gastrointest Endosc. 2020 Sep;92(3):578-588.e4.) The authors should mention it in the discussion. Also, the primary outcome is high risk stigmata, i.e., need for endoscopic treatment, and the cutoffs for GBS and AIMS65 to rule out are both reported as 1. The authors should include this paper in the meta-analysis.

The HARBINGER score was designed to predict the presence of endoscopic high-risk stigmata rather than need for hospital-based intervention. The primary outcome of this meta-analysis is the validated hospital-based outcome rather than endoscopic findings hence HARBINGER did not fulfill the inclusion criteria. Furthermore, in order to be included in the final analysis, we set a prior requirement of the presence of at least 3 validation studies, which the HARBINGER score did not meet as well. These are the main factors why this score was not included.

The authors of the HARBINGER study reported the primary outcome as endoscopic high-risk stigmata. However, they excluded patients who had “oozing” bleeding which is also considered a high-risk lesion in which practice guidelines still recommend endoscopic therapy. Hence, the 2 definitions of high-risk stigmata and need for endoscopic therapy adopted in that study cannot be used interchangeably, and therefore the reported outcomes for GBS and AIMS65 from the HARBINGER study cannot be used to further inform our meta-analysis. 

Reviewer 2 Report

Authors have performed a comprehensive review of studies from different regions which demonstrates diversity and rigor of the techniques adopted. However, there are some points that need to be clarified before concluding on the final decision for the paper.
1.    Authors should outline a proper study question using PICOT framework. At the moment, no such information is elaborated in the mentioned sections.
2.    The data analysis needs to be clarified further. In this section, the authors mentioned that they have applied WMD/SMD or Odd Ratios wherever applied. However, in the results, it is challenging to understand which information is processed using which method. In the result section, e.g. table 3 displays proportional analysis instead of SMD or WMDs
3.    It is challenging to understand the rationale behind OR in Table 4.
4.    In my understanding, Table 3 utilized continuous data and applying analysis on such data is not suitable because of the narrow confidence interval.
5.    Forest plot for table 3 is missing
6.    Data on quality of soundness is missing. Moreover, subgroups analysis is required using QoS so that high quality studies can be segregated.
7.    Study characteristics should be elaborated in detail.

Author Response

1. Authors should outline a proper study question using PICOT framework. At the moment, no such information is elaborated in the mentioned sections.

Population – patients presenting to the ER with suspected upper GI bleeding

Intervention – evaluation of low-risk patient using a pre-endoscopic risk score to predict outcomes

Control – non-low risk patients according to varying thresholds

Outcomes – The primary outcome was a composite score for the need of a hospital-based intervention (endoscopic therapy, surgery, angiography, or blood transfusion). Secondary outcomes included: mortality, rebleeding or the individual endpoints of the composite outcome

T – follow-up up to 30 days from the index bleeding episode

2. The data analysis needs to be clarified further. In this section, the authors mentioned that they have applied WMD/SMD or Odd Ratios wherever applied. However, in the results, it is challenging to understand which information is processed using which method. In the result section, e.g. table 3 displays proportional analysis instead of SMD or WMDs

We have added the heading in table 3 to clarity that the results are presented as proportions (95%CI)

3. It is challenging to understand the rationale behind OR in Table 4.

Using OR helped to compare different scores cut-offs and measuring the accuracy of different scores/cut offs to choose the optimal cut off to identify low-risk population. This is how we identified that GBS score of 0-1 was accurate and optimal for identifying low-risk patients for requiring hospital-based intervention but also a score of up to 2 still had similar accuracy justifying our conclusion that using a GBS cut off of up to 2 is acceptable. We have also added the heading in table 4 to clarity that the results are presented as OR (95%CI)

4. In my understanding, Table 3 utilized continuous data and applying analysis on such data is not suitable because of the narrow confidence interval.

We apologize for the confusion but table 3 lists proportions as is now clearer based on the response to point #2

5. Forest plot for table 3 is missing

The forest plots for proportions have been added.

6. Data on quality of soundness is missing. Moreover, subgroups analysis is required using QoS so that high quality studies can be segregated.

Data quality was characterized using the Ottawa-Newcastle score (NOS) for observational studies. We also performed subgroup analyses based on high quality studies and are presented in the supplementary tables. Should the reviewer require more information, we would require more specific guidance understanding that we are limited by the available data and have no prior experience using quality of soundness for biological research.

7. Study characteristics should be elaborated in detail.

We have added all available study characteristics in the table that include the number of patients with a score above or below the threshold for the given risk score) and the outcome(s) studied in the given publication.

Round 2

Reviewer 1 Report

I felt that the author had no desire to improve the paper in response to the reviewer's recommendations.

Author Response

We are truly sorry by your reaction to our response and want to be clear that we respectfully addressed all suggestions, but we were obviously not clear enough in justifying these. Please allow us to elucidate further to clear up any misunderstanding. Find below our revised responses to your comments and some additional changes made to the manuscript as a result. Hoping you will find these appropriate and acceptable. Thank you.

1. The percentage of each outcome in the composite outcome should be indicated.

Of this composite outcome, the majority may be blood transfusion, which is virtually equivalent to predicting blood transfusion. This point should be discussed.

The individual components of the composite outcome were reported as secondary outcomes when available. In addition, the composite outcome (hospital-based intervention) aims at detecting any intervention which warrants admission regardless of its perceived seriousness. So in case of blood transfusion, even if all the composite outcome was composed entirely of blood transfusion outcome, this would be an indication for admission which is the main aim of the risk assessment scores addressing low-risk patients (i.e.: Identify patients who are unlikely to require hospital-based intervention and be managed as outpatient safely). This choice of composite outcome has been previously validated and endorsed by guidelines, justifying its choice. However, we do understand the underlying clinical rationale for the request. Unfortunately, as we do not have patient-level information from the studies, it is impossible for us to mention which of the patients who met the composite outcome measure experienced each of its individual components. A sentence clarifying this has been added in the discussion section.

2. The sensitivity and specificity for each cutoff value should be summarized. It would also be of interest how many people below that cutoff are included in the total. If the sensitivity is high but the number of people in the population below the cutoff is small, it will not be possible to select many people who do not need treatment.

The aim of this meta-analysis is to compare different thresholds, and thus risk ratios and not absolute test performance characteristics are presented. A meta-analysis of diagnostic tests employs a very different methodology which was not used as this was not the clinical or methodological aim of our meta-analysis. We have included a sentence specifying this consideration in the discussion section.  We have now, as requested, added in the tables the number of patients included for each threshold cut-off analyzed. The only risk score with clinically meaningful thresholds with regards to prognostication of low-risk patients was the GBS, more specifically cut-offs of 1 or 2. We agree completely with the insightful query of reviewer 1 and this is why, in the original submission, we provided the data contrasting the discriminative ability with the overall proportion of patients that such a cut-off would be pertinent to. We thus already have included a discussion for the only risk measure in which the data are sufficient to explore such a trade-off with sufficient statistical power. It is the GBS in which we clearly discuss in the results and the discussion the trade-off of test performance characteristics with the proportion of patients affected by using a threshold score of 1 or 2.

3. No mention is made in the discussion of the recently reported HARBINGER to determine the indication for emergency endoscopy. (Gastrointest Endosc. 2020 Sep;92(3):578-588.e4.) The authors should mention it in the discussion. Also, the primary outcome is high risk stigmata, i.e., need for endoscopic treatment, and the cutoffs for GBS and AIMS65 to rule out are both reported as 1. The authors should include this paper in the meta-analysis.

The HARBINGER score was designed to predict the presence of endoscopic high-risk stigmata rather than the need for hospital-based intervention. Accordingly, the HARBINGER scale is used to predict outcomes amongst high-risk patients. Moreover the performance of an endoscopy is required, and our study assessed patient assessment prior to a gastroscopy since one of the aims is to predict who can have an out-patient gastroscopy performed in a more elective setting. Also, it is important realize that even the absence of such endoscopic high-risk stigmata does not exclude patients with significant co-morbidities or hemodynamic presentation that would not permit them to be considered part of a low-risk group. This scale is thus not pertinent to our submitted meta-analysis, which is why this particular study (like many others assessing high-risk patients) was not included for analysis. In conclusion: The primary outcome of this meta-analysis is the validated hospital-based outcome rather than endoscopic findings hence HARBINGER did not fulfill the inclusion criteria. Furthermore, in order to be included in the final analysis, we set a prior requirement of the presence of at least 3 validation studies, which the HARBINGER score did not meet as well. These are the main reasons why this score was not considered. Also, the authors of the HARBINGER study reported the primary outcome as endoscopic high-risk stigmata. However, they excluded patients who had “oozing” bleeding which is also considered a high-risk lesion in which practice guidelines still recommend endoscopic therapy (Forrest Ib). Hence, even the 2 definitions of high-risk stigmata and need for endoscopic therapy adopted in that study cannot be used interchangeably, and therefore the reported outcomes for GBS and AIMS65 from the HARBINGER study also cannot be used to further inform our meta-analysis. A sentence explaining these considerations has been added to the discussion section as well.